# Effects of Immediate and Delayed Cementations for CAD/CAM Resin Block after Alumina Air Abrasion on Adhesion to Newly Developed Resin Cement

**DOI:** 10.3390/ma14227058

**Published:** 2021-11-21

**Authors:** Akane Chin, Masaomi Ikeda, Tomohiro Takagaki, Toru Nikaido, Alireza Sadr, Yasushi Shimada, Junji Tagami

**Affiliations:** 1Cariology and Operative Dentistry, Graduate School of Medical and Dental Sciences, Tokyo Medical and Dental University (TMDU), 1-5-45 Yushima, Tokyo 113-8549, Japan; chiope@tmd.ac.jp (A.C.); shimada.ope@tmd.ac.jp (Y.S.); tagami.ope@gmail.com (J.T.); 2Oral Prosthetic Engineering, Graduate School, Tokyo Medical and Dental University (TMDU), 1-5-45 Yushima, Tokyo 113-8549, Japan; 3Department of Operative Dentistry, Division of Oral Functional Science and Rehabilitation, School of Dentistry, Asahi University, 1851 Hozumi, Mizuho-City 501-0296, Japan; takagaki@dent.asahi-u.ac.jp (T.T.); nikaido-ope@dent.asahi-u.ac.jp (T.N.); 4Department of Restorative Dentistry, School of Dentistry, University of Washington, 1959 NE Pacific St., Seattle, WA 98195, USA; arsade@uw.edu

**Keywords:** self adhesive resin cement, micro tensile test, CAD/CAM resin composite block, long carbon chain

## Abstract

The purpose of this study was to evaluate the effect of one week of Computer-aided design/Computer-aided manufacturing (CAD/CAM) crown storage on the μTBS between resin cement and CAD/CAM resin composite blocks. The micro-tensile bond strength (μTBS) test groups were divided into 4 conditions. There are two types of CAD/CAM resin composite blocks, namely A block and P block (KATANA Avencia Block and KATANA Avencia P Block, Kuraray Noritake Dental, Tokyo, Japan) and two types of resin cements. Additionally, there are two curing methods (light cure and chemical cure) prior to the μTBS test—Immediate: cementation was performed immediately; Delay: cementation was conducted after one week of storage in air under laboratory conditions. The effect of Immediate and Delayed cementations were evaluated by a μTBS test, surface roughness measurements, light intensity measurements, water sorption measurements and Scanning electron microscope/Energy dispersive X-ray spectrometry (SEM/EDS) analysis. From the results of the μTBS test, we found that Delayed cementation showed significantly lower bond strength than that of Immediate cementation for both resin cements and both curing methods using A block. There was no significant difference between the two types of resin cements or two curing methods. Furthermore, water sorption of A block was significantly higher than that of P block. Within the limitations of this study, alumina air abrasion of CAD/CAM resin composite restorations should be performed immediately before bonding at the chairside to minimize the effect of humidity on bonding.

## 1. Introduction

Technology of computer-aided design and computer-aided manufacturing (CAD/CAM) has remarkably progressed and is becoming popular in medicine, dentistry and industry. Glass ceramics, zirconia and resin composites are the materials used for the fabrication of dental restorative and prosthetic devices based on CAD/CAM technology in dentistry. In recent years, CAD/CAM technology has been utilized to fabricate indirect restorations from CAD/CAM resin composite blocks (CRBs) with differing compositions. CRBs are industrially polymerized and are expected to have an even better polymerization ratio in comparison to laboratory-fabricated indirect resin composite [1,2,3,4,5,6].

The CRBs can be manufactured using different processes; one common process involves the polymerization of a resin composite paste made of the matrix monomers and inorganic fillers. In a different method, inorganic filler particles are compressed with a high pressure and then infiltrated with a matrix monomer, after which the resin is polymerized to produce the CRB [7]. The mechanical properties of CRBs improved under polymerization conditions of a higher temperature and/or a higher pressure [8].

The CAD/CAM technology in combination with adhesive dentistry creates an opportunity to take advantage of minimally invasive preparation designs, which save the natural tooth structure. Therefore, the role of adhesive resin cement is crucial to the success of CAD/CAM resin composite restorations [9,10]. The bonding performance of dental adhesives is commonly evaluated by the micro-tensile bond strength (μTBS) test [9]. There are few studies that have reported the μTBS between CRBs and the resin cement [9,11,12,13]. The air abrasion of resin composite intaglio was recommended to increase roughness and adhesive surface area, which would result in increased mechanical retention between the resin composite and the adhesive material [14]. The CRBs contain some filler particles; therefore, it was suggested that application of functional monomers (such as silane coupling agents) that could chemically bond to the fillers improved bonding to the CRB in addition to air abrasion [10].

The CRB milling and air-abrasion procedure may be performed in a dental laboratory, and the restoration may be kept for several days prior to the restoration delivery appointment. The surface of the CAD/CAM crown would absorb moisture from the air during the storage period after air abrasion. It was reported that the adhesion of the resin cement was reduced, and the bonding strength decreased after storage [9,14]. For this reason, chairside air abrasion was recommended immediately before setting. However, a chairside CAD/CAM and/or an air-abrasion unit may not be available in all dental clinics, and optimal storage conditions of the restoration prior to delivery should be determined.

Therefore, the purpose of this study was to evaluate the effect of one-week storage on the bond strength. Two types of CAD/CAM resin composite blocks with different compositions, namely A block and P block (KATANA Avencia Block and KATANA Avencia P Block, Kuraray Noritake Dental, Tokyo, Japan), were used in this study. A block is recommended to fabricate premolar restorations and P block for molar restorations, and there is a difference in the Silicate filler content between two different CAD/CAM resin composite blocks. In addition, two types of resin cement, namely Plus and Multi (SA Luting Plus and SA Luting Multi, Kuraray Noritake Dental, Tokyo, Japan), were used to evaluate the effect of different silanization procedures on bond strength in this experiment. Multi is self-adhesive resin cement that included silane in its paste composition, while SA Luting Plus is a conventional self-adhesive resin cement, which would require a separate silanization step for bonding to glass-based substrates.

The null hypotheses were that the types of CAD/CAM resin composite blocks, storage condition and silanization procedure on the adhesive surface would not affect the bond strength between self-adhesive resin cement and CRBs, and there were no differences in bond strength to CRBs between newly developed self-adhesive resin cement containing a silane coupling agent and the conventional resin cement with silane treatment.

## 2. Materials and Methods

The flowchart of the all study is shown in Figure 1. And the materials, product names, manufacturers, application procedures and compositions used in this study are listed in Table 1.

### 2.1. Micro Tensile Bond Strength Test (μTBS)

The CRBs were sectioned with a diamond disk (IsoMet, 11-1280-170 Buehler, Lake Bluff, IL, USA) to obtain 2 mm thick slices (A block; 10.5 × 12.5 × 2 mm, P block; 14.5 × 12.5 × 2 mm). Each slice was then ground by #600 SiC paper (Water proof Abrasive Paper Sheet, Sankyo Rikagaku, Tokyo, Japan) in wet conditions. After that, the specimens were treated by air-particle abrasion with 50-μm Al_2_ O_3_ particles (Cobra 1594-1205 50μ, Renfert, Hilzingen, Germany) at 0.2 MPa (20 s, 10 mm distance) using an air-abrasion unit (Basic Master, Renfert, Hilzingen, Germany). All specimens were cleaned with 99% ethanol for 3 min in an ultrasonic bath (US-2KS SND Corporation, Nagano, Japan).

The specimens were then divided into two subgroups according to the time passed after alumina air abrasion and prior to the μTBS test—Immediate: cementation was performed immediately; Delay: cementation was conducted after one week of storage in air under laboratory conditions, where the temperature was set to 23.0 ± 0.5 °C (room temperature) and the relative humidity (RH) was 50 ± 5% [9]. Then, all CRB surfaces were etched by phosphoric acid (K-etchant gel, Kuraray Noritake Dental, Tokyo, Japan) for 20 s, cleaned with distilled water and air-dried.

For specimens in the Plus group, a silane coupling agent (Clearfil Ceramic Primer Plus, Kuraray Noritake Dental, Tokyo, Japan) was applied to the adhesive surface of CRB and air-dried. Conversely, for those in Multi, the bonding was performed directly to the adhesive surface of CRB without silane treatment.

Before bonding, 100 μm thick aluminum tape was placed on the adhesive surface to control cement thickness. Then, both types of cement were applied to the CRB, and the blocks were attached to each other. The specimens were polymerized in two subgroups: the chemical cure group (CC), in which the specimens were left in the dark for 30 min without light irradiation, and the light-cure group (LC), in which they were irradiated with an LED light curing unit (Valo LED curing light, Ultradent, South Jordan, UT, USA) for a total of 40 s and 20 s, each on the top and bottom surfaces.

After the specimens were stored for 24 h in distilled water, µTBS samples (1 × 1 mm beams) were prepared with the diamond disk (IsoMet, 11-1280-170 Buehler, Lake Bluff, IL, USA) at the adhesive–resin cement interface. A total of 20 beams were prepared in each subgroup. The beams were carefully attached to a test jig with a cyanoacrylate glue (Model Repair II Blue, DentsplySankin, Tokyo, Japan), and then the μTBS test was conducted using a universal testing machine (EZ-Test, Shimadzu, Kyoto, Japan) at a crosshead speed of 1 mm/min [9].

### 2.2. Failure Mode Analysis

After the μTBS test, a total of 20 fracture beams in each subgroup were placed on a carbon adhesive tape (Nisshin EM, Tokyo, Japan) on the specimen stages and fixed with cyanoacrylate glue (Model Repair II Blue, DentsplySankin, Tokyo, Japan) for scanning electron microscopy (SEM; JSM5310LV, JEOL, Tokyo, Japan) observation of the separated interface. The beams were then gold sputter-coated and inspected by SEM at 100× magnification [9]. The failure modes were classified into two categories: (A) adhesive failure between CRB and resin cement and (B) cohesive failure in resin cement.

### 2.3. Scanning Electron Microscope/Energy Dispersive X-ray Spectrometry (SEM/EDS) Analysis

Five slices from each block were made using the diamond disk (IsoMet, 11-1280-170 Buehler, Lake Bluff, IL, USA). The surface composition of A block and P block after alumina air abrasion and cleaning with ethanol was observed by SEM (Feg-SEM; JSM-6701F, JEOL, Tokyo, Japan). The thin layer of carbon was coated on the specimens using a JEE-420T vacuum deposition system (JEE-420T Vacuum Evaporators, JEOL, Tokyo, Japan) and then observed with field-emission-gun SEM (Feg-SEM; JSM-6701F, JEOL, Tokyo, Japan) at 5 kV with an annular semiconductor detector. The determined area was 0.06 mm × 0.05 mm for each material.

### 2.4. Light Intensity Measurements (mW/cm^2^)

The control light intensity value was measured three times with the light curing unit tip (Valo LED curing light, Ultradent, South Jordan, UT, USA) at a distance of 0 mm from the visible light radiometer (CURE RITE, DENTPLY Caulk, Dentsply Sirona, Bensheim, Germany), and the average light intensity was used as the representative control value. To determine attenuation through each CRB material, 9 slices each of A block and P block were cut with the diamond saw (IsoMet, 11-1280-170 Buehler, Lake Bluff, IL, USA) at the same thickness as μTBS test specimens. After alumina air abrasion, the samples were cleaned with ethanol, and the light intensity was measured three times on the bottom surface of the prepared slice using the light intensity meter.

### 2.5. Surface Roughness Measurements (Sa)

In order to evaluate surface roughness (Sa) of the A block and P block, 9 slices (10.0 × 10.0 × 2.0 mm) of A block and P block were cut with a diamond saw (IsoMet, 11-1280-170 Buehler, Lake Bluff, IL, USA). After alumina air abrasion, cleaning with ethanol and etching by phosphoric acid, 3 areas, each 1.5 mm in diameter, were randomly selected at the center of the prepared slice and analyzed using a confocal laser scanning microscope (CLSM) (Keyence VK-X150, Keyence Corp., Osaka, Japan). The surface roughness (Sa) was calculated by averaging the surface roughness of the 3 areas in each specimen.

### 2.6. Water Sorption Measurements

As for water sorption measurements, 5 slices (10.0 × 10.0 × 2.0 mm) of A block and P block were prepared using the diamond saw (IsoMet, 11-1280-170 Buehler, Lake Bluff, IL, USA). After alumina air abrasion, the samples were cleaned with ethanol and stored under laboratory conditions (room temperature: 23.0 ± 0.5 °C, relative humidity: 50.0 ± 5.0 °C) for 1 day, 3 days, 5 days and 1 week) [9]. All specimens were measured by weight using analytical balance (AUW120D, Shimadzu, Kyoto, Japan) at each period. The water sorption ratio was then calculated from the following formula:A = 100(Wn − W0)/W0
where α is water sorption (%), Wn is the weight of the specimen at each measurement period (in mg) and W0 is the weight of the specimen immediately after it was cleaned with 99% ethanol for 3 min in an ultrasonic bath (US-2KS SND Corporation, Nagano, Japan).

### 2.7. Statistical Analysis

Normality and equal variances of data were analyzed by Shapiro–Wilk test and Levene’s test, respectively. An appropriate test was used for analysis that considered both the normality and variance of the results. In this manner, the groups were compared by t-test for surface roughness (n = 9) and water sorption (n = 5), *t*-test (Welch method) with Bonferroni correction for light intensity (n = 9) and micro tensile bond strength (n = 20), and Chi-square test and Fisher exact test with Bonferroni correction for the frequency of failure mode (n = 20). The significance level was set at 0.05, and Power was set at 80%. The sample size was initially calculated from the pilot study.
$$n=\frac{2*{\left(1.96+0.84\right)}^{2}{*(\mathrm{SD})}^{2}}{{{(\mathrm{Av}}_{1}-{\mathrm{Av}}_{2})}^{2}}$$
n: number of specimens in each experimental group, SD: standard deviations. Av_1_ and Av_2_: Average value in each experiment.

All Statistical analysis was performed using Statistical software (SPSS ver. 26.0 for Windows, IBM Corp., Armonk, NY, USA).

## 3. Results

### 3.1. Micro Tensile Bond Strength (μTBS)

The mean μTBS values and frequency in mode of failure were shown in Table 2.

There was no significant difference between CC and LC except for the P block group with resin cement Plus in the Immediate group (*p* > 0.05). In the CC group, there was significant difference between A block and P block in each resin cement in the Delay group (*p* < 0.05). In LC groups, there was significant difference between A block and P block using Multi resin cement in the Delay group (*p* < 0.05). There was no significant difference between CC and LC except in the Immediate group with Plus resin cement group and P block. In both chemical and light cure modes of A block, there was a significant difference between Immediate cementation and Delayed cementation for each resin cement (*p* < 0.05). There was no significant difference between the two resin cements (*p* > 0.05). The frequency of adhesive failures for the Delay group indicated significant increase over that of the Immediate group in each resin cement for both A and P blocks (*p* < 0.05). There was no significant difference between resin cements, CAD/CAM resin composite blocks and curing conditions (*p* > 0.05).

### 3.2. SEM/EDS Analysis

The results of the typical SEM/EDS analysis were shown in Figure 2 and Table 3. Nano-fillers were observed on the surface of A block (Figure 2A). However, on the surface of P block (Figure 2B), both nano-fillers and micro-irregular fillers were exposed. The elemental mapping results are shown in Table 3; in the elemental analysis, Si was observed on both the surface of A block and P block, while Ba was observed on the surface of P block only.

### 3.3. Light Intensity Measurements (mW/cm^2^)

The mean light intensity values are shown in Table 4. There were significant differences among Control, A block and P block groups. The light intensity of the control group was about 678 (mW/cm^2^), while the light intensity of the A block group was about 19 (mW/cm^2^) and the P block group was about 71 (mW/cm^2^), which are significantly lower than that of control group (*p* < 0.05).

### 3.4. Surface Roughness Measurements (Sa)

The mean surface roughness values after aluminum air abrasion are shown in Table 5. Surface roughness (Sa) was 1.47 for of A block and 1.37 for P block; there was no significant difference between A and P groups after aluminum air abrasion (*p* > 0.05).

### 3.5. Water Sorption Measurements

The mean water sorption values in each storage period are shown in Table 6. After 1 week, water sorption (%) of both A block and P block groups were significantly higher than that after 1 day of storage, and there was significant difference between A block group and P block group in each storage period.

## 4. Discussion

CRBs designated for anterior teeth, molars and premolars may have different properties. The CRB recommended by the manufacturer for larger (molar) restorations (P block) is characterized by a greater filler/matrix monomer ratio than that of the CRB for premolars (A block). Mechanical properties of the CRBs, such as flexural strength, Vickers hardness and compressive strength, are influenced by filler content ratio and size, comparable to resin composite materials recommended for direct/indirect restorations [17,18,19,20,21,22]. The P block contained a high content of filler particle (approximately 82 wt%) in comparison with approximately 62 wt% in A block, which is a relatively low content [9,16].

Some self-adhesive resin cements were used for the cementation of CRB [23,24]. The intaglio surface of the CRB restoration is commonly treated with a silane coupling agent before cementation. SA Luting Multi has a new composition that contained a silane coupling agent in the paste, so the manufacturer was does not recommend the use of any silane primer before cementation with this material. In this experiment, it was hypothesized that the direct application of a silane primer would provide a higher bond strength than using the newly developed self-adhesive resin cement containing a silane coupling agent; previous studies have recommended multi-step procedures improve the performance of adhesives [25,26]. However, the results of the µTBS showed no difference in the bond strength of the two types of resin cement.

It was reported that γ-Mercaptopropyltrimethoxysilane (γ-MPTS) could be easily hydrolyzed in an aqueous acidic adhesive formula. The newly developed cement contained a silane coupling monomer that has a longer carbon chain than γ-MPTS according to the manufacturer (https://www.trademarkelite.com/europe/trademark/trademark-detail/017996211/Chemical-Affinity-Long-Carbon-chain-Silane-Coupling-Agent-LCS), (11 November 2021), which is more hydrophobic. This monomer did not show hydrolysis when incorporated into the resin cement paste, which could explain why no difference was found in µTBS between the two types of self-adhesive resin cement in this study of [27].

On the other hand, the intaglio surface of the CRB restoration is abraded by air abrasion before restoration delivery with resin cement. It has also been reported that the alumina particle size of 50 μm is effective to provide micromechanical retention for bonding to polymerized resin composite [14]. In terms of air-abrasion pressure, 0.1 or 0.2 MPa have been commonly recommended; higher pressures, such as 0.4 MPa, increase the mechanical retention with the resin cement but may also damage the CRB surface and cause chipping [9,28]. Therefore, in this study, specimens were treated with 50 μm aluminum oxide particles at 0.2 MPa. It is well known that air-abrasion treatment increases the bond strength of restorations with the resin cement [13,28]. In the present study, there was no difference in Sa between A block and P block; therefore, it can be assumed that mechanical retention and surface area did not affect the results of the μTBS test.

In this study, we also measured the light intensity to investigate the difference in light attenuation between A block and P block. The thickness of specimens in the light intensity measurement was 2 mm same thickness as µTBS specimens because minimum thickness is 1.5 mm based on the instructions (https://katanaavencia.com/wp-content/uploads/KATANA_AVENCIA_Block_IFU.pdf), (11 November 2021). A relationship among the bond strength, irradiation distance and light intensity has been shown for a long time [29,30,31]. The lower the light intensity for adhesive curing, the lower the bond strength will be [29,30,31]. The results of light intensity measurements revealed that the light intensity of P block was significantly higher than that of A block, which may be explained by the difference of light diffusion properties between blocks. However, the results showed that the light intensity passing through 2 mm thick specimen of both A block and P block was very low. Furthermore, from the results of µTBS, chemical cure (CC) and light cure (LC) results were almost the same for all groups. This suggests that the resin cement was mainly polymerized by chemical curing.

The study results suggested a relationship between storage conditions and µTBS. The CRB restorations may be stored for a long time in a dental laboratory or clinic after fabrication. It was reported that water immersion might cause a reduction in bond strength on the adhesive surface, but there are few reports on the effect of moisture in the air. In this study, we compared the bond strength and failure mode due to air humidity. From the results of bond strength, we found that Immediate cementation showed significantly higher bond strength than that of Delayed cementation for both types of resin cement and both curing methods using A block. However, using the P block, no difference was observed between Immediate and Delayed cementations. Furthermore, from the result of failure mode analysis, the number of adhesive failures of Delayed cementation was observed more than that of Immediate cementation. In addition, from the result of water sorption measurements, it was demonstrated that A block absorbed moisture more than P block. Considering the findings of surface roughness, light intensity, water sorption and µTBS experiments under different storage conditions, it was postulated that adhesion to CRB was most influenced by moisture in the air in this experiment.

The filler content, particle size and shape are the differences between A block and P block used in this experiment. A block contained 62 wt% of filler particle and P block contained 82 wt% in this experiment. It was initially thought that A block would provide a significantly higher bond strength than P block because there is a smaller amount of Si and little water sorption from the air. However, the bond strength of A block decreased, whereas that of P block did not decrease after 1 week of storage. It was reported that adhesion between the filler and matrix monomer was dependent on both the filler concentration and particle size in the resin composite material [32,33,34]. In addition, the bond strength between the resin cement and the resin composite containing nano-filler significantly decreased after 2 weeks of water storage, while there was no significant difference in that for resin composite containing micro-filler [35]. Moreover, silane treatment for micro-particle size filler is more effective than that treatment for nano-particle size filler [35]. Nevertheless, it was reported that Ba-glass and the adhesive interface between Ba-glass filler and matrix monomer were influenced by water sorption differently from Si filler, where the water sorption rate decreased with the bigger Ba-glass filler size [34]. The SEM/EDS observation showed exposed filler particles in both A block and P block after air aluminum abrasion. Nano-filler cluster formation was in A block, whereas both micro-irregular and nano-filler particles were observed on the surface of P block. These differences could justify the reason why unlike A block, there was no significant difference in μTBS between the Immediate and Delayed cementations of P block; a high bonding performance was achieved from the silane coupling agent in both resin cement systems, and Ba-glass filler was less influenced by storage humidity, as previously reported [34,35].

The differences between A block and P block in filler content, size and type resulted in the differences in the effects from moisture in the air, which was reflected in the results of the μTBS in this study. Therefore, compositional factors affected the result of μTBS, but it was not possible to determine which factor had the greatest effect. In addition, there were no statistically significant difference between two types of resin cements in μTBS in this study. However, various factors for cementation may affect the results of μTBS using other resin cement. It is suggested that future studies for CAD/CAM resin composite blocks compare the results with controlled filler contents, sizes, types and other types of resin cements.

## 5. Conclusions

Within the limitations of this study, the newly developed silane-containing resin cement was effective in bonding to CRBs. Storage air humidity affected bonding to CRB after aluminum air abrasion, and the effect depended on the composition of the CAD/CAM resin composite block. Air abrasion of CRB restorations should be performed immediately before bonding at the chairside to minimize the effect of humidity on bonding.

## Figures and Tables

**Figure 1 materials-14-07058-f001:**
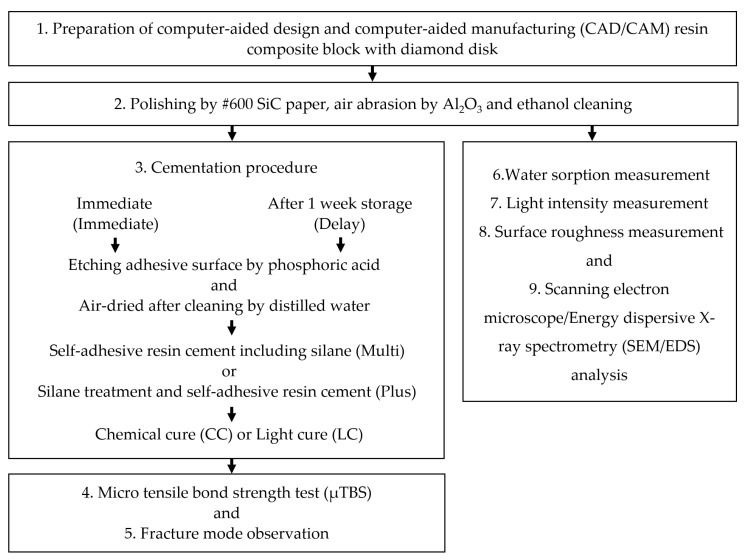
The flowchart of the all study.

**Figure 2 materials-14-07058-f002:**
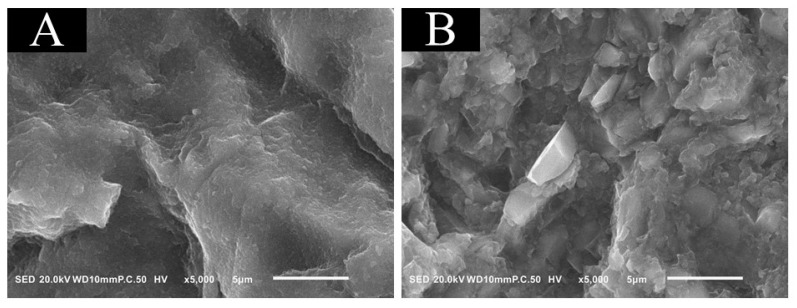
Results of the typical SEM analysis. SEM observation of CRB surface at 5000× magnification after alumina air abrasion at 0.2 MPa: A block (**A**) and P block (**B**). Many nano-dimples were observed on the surface of A block, and many micro-irregular dimples were observed on the surface of P block.

**Table 1 materials-14-07058-t001:** Composition of the materials used in the study and their applications procedures.

Material	Lot No.	Composition	Application Procedures
KATANA Avencia Block	000402	Mixed filler with colloidal silica (40 nm) and aluminum oxide (20 nm), cured resins consisting of methacrylate monomer (Copolymer of UDMA and other methacrylate monomers), pigments, filler content 62 (wt%) *	
KATANA Avencia P Block	000130	barium glass filler, silica glass filler, UDMA, pigments, others, filler content 82 (wt%) *	
SA Luting Plus	450163	Bis-GMA, TEGDMA, methacrylic acid type monomer, MDP, barium glass, silica type microfiller photopolymerization catalyst, chemical polymerization catalyst, surface treated sodium fluoride *	Apply it by auto-mix syringe on the CRB surface, then light cure for 40 s or chemical cure for 30 min in the dark.
SA Luting Multi	T180219	MDP, Bis-GMA, TEGDMA, HEMA, silica glass filler, hydrophobic methacrylic acid monomer, barium glass filler, Aluminum oxide, NAF, Newly developed silane coupling agent *	Apply it by auto-mix syringe on the CRB surface, then light cure for 40 s or chemical cure for 30 min in the dark.
K-etchant gel	4Q0078	Water, 40%phosphoric acid, pigment, thickener *	Apply on the CRB surface for 20 s, rinse with water for 10 s and air-dry gently.
Clearfil Ceramic Primer Plus	A50030	Silane coupling agent γ-MPTS, MDP, ethanol *	Apply on the CRB for 20 s and air-dry gently

Manufacturer: Kuraray Noritake Dental, Tokyo, Japan. 10-MDP: 10-methacryloyloxydecyl dihydrogen phosphate; HEMA: 2-hydroxyethyl methacrylate; Bis-GMA: bisphenol-Adiglycidyl methacrylate; TEGDMA: triethyleneglycol dimethacrylate. * References [9,15,16].

**Table 2 materials-14-07058-t002:** Results of μTBS and frequency in mode of failure.

Curing Methods	CAD/CAM Resin Composite Blocks	Cementations	Resin Cements	μTBS	Frequency in Mode of Failure (A/B)
CC	A block	Immediate	Multi	73.52 ^a^ (6.49)	(0/20 ^a^)
			Plus	70.19 ^b^ (8.87)	(0/20 ^b^)
		Delay	Multi	43.46 ^a,c^ (12.91)	(20/0 ^a^)
			Plus	51.03 ^b,d^ (7.56)	(19/1 ^b^)
	P block	Immediate	Multi	64.33 (7.52)	(4/16 ^c^)
			Plus	60.86 ^e^ (7.19)	(0/20 ^d^)
		Delay	Multi	61.21 ^c^ (9.14)	(20/0 ^c^)
			Plus	65.2 ^d^ (7.24)	(18/2 ^d^)
LC	A block	Immediate	Multi	68.48 ^f^ (6.74)	(2/18 ^e^)
			Plus	69.77 ^g^ (9.8)	(3/17 ^f^)
		Delay	Multi	42.31 ^f,h^ (6.36)	(20/0 ^e^)
			Plus	42.24 ^g^ (6.55)	(20/0 ^f^)
	P block	Immediate	Multi	68.05 (8.49)	(7/13 ^g^)
			Plus	71.53 ^e^ (11.89)	(3/17 ^h^)
		Delay	Multi	64.39 ^h^ (8.29)	(20/0 ^g^)
			Plus	63.64 (7.42)	(19/1 ^h^)

Data are shown as mean (standard deviation). Numbers in square brackets are the number of specimens classified into two fracture modes (A/B). A: adhesive failure, B: cohesive failure in resin cement. Same letters indicate statistically significant differences in each column (n = 20, *p* < 0.05). CC: Chemical cure group, LC: Light cure group. CAD/CAM: Computer-aided design/Computer-aided manufacturing.

**Table 3 materials-14-07058-t003:** Results of EDS analysis on the surface of CAD/CAM resin composite blocks after alumina air abrasion.

	CAD/CAM Resin Composite Blocks	Elements Formula					
		C	O	Al	Si	Ba	Toatl
Atom (%)	A block	44.95	40.38	0.31	14.36	0.00	100.00
	P block	33.58	49.65	2.93	11.47	2.37	100.00
mass (%)	A block	33.79	40.44	0.52	25.24	0.00	100.00
	P block	20.97	41.29	4.11	16.74	16.88	100.00

**Table 4 materials-14-07058-t004:** Results of light intensity measurements.

CAD/CAM Resin Composite Blocks	Light Intensity (mW/cm^2^)
A block	19.22 (4.06) ^A^
P block	71.00 (4.36) ^A^
Control	677.89 (4.96) ^A^

Data are shown as mean (standard deviation). Control: light irradiator at a distance of 0 mm was measured. Same uppercase letters indicate statistically significant differences (n = 9, *p* < 0.05).

**Table 5 materials-14-07058-t005:** Results of surface roughness measurements.

CAD/CAM Resin Composite Blocks	Surface Roughness (Sa)
A block	1.37 (0.09)
P block	1.47 (0.12)

Data are shown as mean (standard deviation). A block: CAD/CAM resin composite block for premolar (KATANA Avencia Block, Kuraray Noritake Dental, Tokyo, Japan). P block: CAD/CAM resin composite block for molar (KATANA Avencia P Block, Kuraray Noritake Dental, Tokyo, Japan).

**Table 6 materials-14-07058-t006:** Result of water sorption measurements.

CAD/CAM Resin Composite Blocks	Water Sorption (%)			
	1 Day	3 Days	5 Days	1 Week
A block	0.07 (0.01) ^a,A^	0.11 (0.01) ^b,A^	0.19 (0.02) ^c,A^	0.27 (0.03) ^d,A^
P block	0.03 (0.01) ^a,B^	0.05 (0.01) ^b,B^	0.09 (0.01) ^c,B^	0.12 (0.01) ^d,B^

Data are shown as mean (standard deviation). Same small letters indicate statistically significant differences in each column. Same large letters indicate statistically significant differences in each low (n = 5, *p* < 0.05). A block: CAD/CAM resin composite block for premolar (KATANA Avencia Block, Kuraray Noritake Dental, Tokyo, Japan). P block: CAD/CAM resin composite block for molar (KATANA Avencia P Block, Kuraray Noritake Dental, Tokyo, Japan).

## Data Availability

Not applicable.

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
