# Peer review of "Effects of Immediate and Delayed Cementations for CAD/CAM Resin Block after Alumina Air Abrasion on Adhesion to Newly Developed Resin Cement"

_materials, 2021, doi:10.3390/ma14227058_

Round 1

Reviewer 1 Report

Dear Authors,

Thank you for this research, I have some general comments wishing they might improve the quality of the manuscript.

The title needs to be changed to reflect what has been done. Instead of writing dry and wet storage which is miss leading terms, maybe it is better to say immediate and delay cementation after air abrasion.

There is a lot of information missing in the abstract part (M & M)

Figure 1 needs to be re-evaluated and it should go in line with the text, which has been done first, and what was the last.

In Table was 1, K-etchant gel was used, but wondering at which stage, because I could not find it from the text.

Throughout the manuscript, choose one term either resin composite or composite resin. Same for other terms. How many samples per group and also sub-groups were not clear in the text.

It looks like you made some characterization of CAD-CAM composite blocks in addition to testing the effect of immediate or delayed cementation after air abrasion. This was not mentioned at all in the abstract nor in the introduction part.

Figure 3 does not add any value and can be replaced by a small table where you show the % of different ions

You need to specify the thickness of CAD-CAM SPECIMENS for Light intensity measurement. The difference between the blocks was huge and this was not probably explained in the discussion part.

The differences between the two types of resin cement could be a result of many reasons and not only silane.

The style of writing the manuscript is not the same as the journal style

Author Response

Dear. Reviewer,

Thank you very much for your revision. We are really grateful for your comments. We would appreciate it if you would read the modified manuscript and our answer relevant it. We would like to answer your question from number 1 to 11.

  1. The title needs to be changed to reflect what has been done. Instead of writing dry and wet storage which is miss leading terms, maybe it is better to say immediate and delay cementation after air abrasion.

Thank you for kind suggestion.

As you mentioned, I changed the title of manuscript for better understand.

Dry and Wet were replaced with immediate and delayed cementations

Title, Figure and text has been modified.

  1. There is a lot of information missing in the abstract part (M & M)

Thank you for your comment.

We add descriptions of water sorption measurements, light intensity measurements and SEM/EDS analysis which were omitted in the abstract.

  1. Figure 1 needs to be re-evaluated and it should go in line with the text, which has been done first, and what was the last.

Thank you for your comment.

Text was divided Micro tensile bond strength test from other measurement test in Figure 1.

  1. In Table was 1, K-etchant gel was used, but wondering at which stage, because I could not find it from the text.

Thank you for your comment.

We apologize for this error. K-etchant gel (phosphoric acid) has been added in Figure 1.and manuscript Line from 147 to 149.

  1. Throughout the manuscript, choose one term either resin composite or composite resin. Same for other terms.

Thank you for your comment.

Composite resin block was replaced with CAD/CAM resin composite block in manuscript. And composite resin was replaced with resin composite.

  1. How many samples per group and also sub-groups were not clear in the text.

Thank you for your kind suggestion.

It was difficult to find the number of specimens just by writing them in each topic, so I wrote the sample numbers for all groups in the statistical analysis topic.

  1. It looks like you made some characterization of CAD-CAM composite blocks in addition to testing the effect of immediate or delayed cementation after air abrasion. This was not mentioned at all in the abstract nor in the introduction part.

Thank you for your comment.

Effect of Immediate or delayed cementation were mentioned in this manuscript.

Descriptions of water sorption measurements, light intensity measurements and SEM/EDS analysis has been added in the abstract.

  1. Figure 3 does not add any value and can be replaced by a small table where you show the % of different ions

Thank you for your comment.

Figure 3 has been modified to Table 3.

  1. You need to specify the thickness of CAD-CAM SPECIMENS for Light intensity measurement. The difference between the blocks was huge and this was not probably explained in the discussion part.

Thank you for your comment.

The difference between A block and P block has been added in Line from 372 to 375 and Line from 378 to 381.

  1. The differences between the two types of resin cement could be a result of many reasons and not only silane.

Thank you for your comment.

There was no significant difference between the cements in this study, but in future studies using other cements, result of test may be affected by a variety of factors.

So, we add description in Line from 425 to 430 in end of discussion part.

  1. The style of writing the manuscript is not the same as the journal style

Thank you for your pointing it out.

The style of writing the manuscript has been slightly modified same style as journal.

Reviewer 2 Report

I enjoyed reviewing your article. It is clear and addresses an important clinical problem.

Abstract: Please avoid abbreviations (CRB) to make the abstract as clear as possible

Introduction: I recommend more lit review to cover the cements used for the study and how silane is hypothesized to make a difference.

Methods and materials: please provide your reference for materials compositions listed in Table 1.

Remaining parts of the study is acceptable with no comments.

All the best

Author Response

Dear. Reviewer,

Thank you very much for your revision. We are really grateful for your comments. We would appreciate it if you would read the modified manuscript and our answer relevant to it. We would like to answer your question from number 1 to 3.

  1. Abstract: Please avoid abbreviations (CRB) to make the abstract as clear as possible.

Thank you for your comment.

CRB was replaced with “CAD/CAM resin composite blocks” in abstract.

  1. Introduction: I recommend more lit review to cover the cements used for the study and how silane is hypothesized to make a difference.

Thank you for your comment.

The hypothesis about silane was added Line 82 in introduction.

  1. Methods and materials: please provide your reference for materials compositions listed in Table 1.

Thank you for your comment.

References No. 9, 15 and 16 were added in Table 1.

Round 2

Reviewer 1 Report

Thank you for the revision and all the best for future research.

References writing style still need to be corrected according to the Journal instructions.

Author Response

Dear. Reviewer,

Thank you very much for your revision. We are really grateful for your comments. We would appreciate it if you would read the modified manuscript

1 References writing style still need to be corrected according to the Journal instructions.

Thank you for your pointing it out. We have modified the writing style of the reference section.

Beast regards,

Masaomi Ikeda